# Modeling transfer of vaginal microbiota from mother to infant in early life

Martin Steen Mortensen[1†], Morten Arendt Rasmussen[2,3†], Jakob Stokholm[2], Asker Daniel Brejnrod[1], Christina Balle[1], Jonathan Thorsen[2], Karen Angeliki Krogfelt[4,5], Hans Bisgaard[2], Søren Johannes Sørensen[1*]

[1]Section of Microbiology, Department of Biology, University of Copenhagen, Copenhagen, Denmark; [2]Copenhagen Prospective Studies on Asthma in Childhood, Faculty of Health Sciences, University of Copenhagen, Copenhagen University Hospital Gentofte, Hellerup, Denmark; [3]Department of Food Science, Faculty of Science, University of Copenhagen, Frederiksberg, Denmark; [4]Virus & Microbiological Special Diagnostics, Statens Serum Institut, Copenhagen, Denmark; [5]Department of Science and Environment, Roskilde University, Roskilde, Denmark

**Abstract** Early-life microbiota has been linked to the development of chronic inflammatory diseases. It has been hypothesized that maternal vaginal microbiota is an important initial seeding source and therefore might have lifelong effects on disease risk. To understand maternal vaginal microbiota's role in seeding the child's microbiota and the extent of delivery mode-dependent transmission, we studied 665 mother–child dyads from the COPSAC$_{2010}$ cohort. The maternal vaginal microbiota was evaluated twice in the third trimester and compared with the children's fecal (at 1 week, 1 month, and 1 year of age) and airway microbiota (at 1 week, 1 month, and 3 months). Based on the concept of weighted transfer ratios (WTRs), we have identified bacterial orders for which the WTR displays patterns indicate persistent or transient transfer from the maternal vaginal microbiome, as well as orders that are shared at later time points independent of delivery mode, indicating a common reservoir.

*For correspondence:
sjs@bio.ku.dk

†These authors contributed equally to this work

Competing interests: The authors declare that no competing interests exist.

## Introduction

Recent studies have suggested that transfer of bacteria from mother to infant during vaginal birth (*Bokulich et al., 2016*; *Dominguez-Bello et al., 2016*) is fundamental for the formation of the early infant microbiota and later disease risk: (1) Delivery mode affects the development of the microbiota in early life, and differences between the microbiota of infants delivered vaginally and by cesarean section (CS) have been identified as late as at 1 year of age (*Azad et al., 2013*; *Bäckhed et al., 2015*; *Stokholm et al., 2016*). (2) Such bacterial colonization in early life has been correlated with the risk of several chronic inflammatory disorders (*Bisgaard et al., 2007*; *Bisgaard et al., 2008*; *Bisgaard et al., 2011*; *Stokholm et al., 2018*; *Wang et al., 2008*), and (3) correlations have been observed between CS and increased risk of such diseases (*Mueller et al., 2017*; *Sevelsted et al., 2015*). Any causal relationship explaining these correlations has yet to be identified, but the consistent observational evidence suggests that some form of causal relationship does exist.

The vaginal microbiota of pregnant women, in comparison with non-pregnant women, has decreased diversity and increased stability (*Aagaard et al., 2012*; *Dominguez-Bello et al., 2010*; *Hernández-Rodríguez et al., 2011*; *Romero et al., 2014*; *Walther-António et al., 2014*), until shortly before delivery (*Rasmussen et al., 2020*), potentially lowering the risk of bacterial perturbations implicated in adverse pregnancy outcomes, including preterm delivery and low birth weight (*Carey et al., 2005*; *Genc et al., 2004*; *Hillier et al., 1995*; *Nadisauskiene et al., 1995*), as well as bacterial vaginosis (*Hummelen et al., 2010*). Several studies of the vaginal microbiota have

identified five vaginal community state types (CSTs): four that are dominated by a specific *Lactobacillus* species and have low alpha diversity, as well as one containing facultative and strictly anaerobic bacteria, with higher alpha diversity, dominated either by *Gardnerella spp.* or by *Gardnerella spp.* with a higher amount of *Lactobacillus spp.* (*Chaban et al., 2014*; *Drell et al., 2013*; *Gajer et al., 2012*; *Hyman et al., 2005*; *Ling et al., 2010*; *Ravel et al., 2011*; *Ravel et al., 2013*; *Zhou et al., 2004*; *Zhou et al., 2007*).

While the very first microbial exposure is dictated by delivery mode – mothers' microbiota during vaginal birth (vaginal and fecal) or skin microbiota after birth by CS (*Dominguez-Bello et al., 2010*; *Chu et al., 2017*), the hypothesis of vaginal seeding has been questioned by *Wampach et al., 2017*, which did not observe a difference between the microbiota of vaginally and CS-delivered neonates before they were 5 days old. Infants' airway or fecal microbiota is naturally very different from the vaginal microbiota (*Dominguez-Bello et al., 2016*; *Stokholm et al., 2018*; *Mortensen et al., 2016*), and the early development is mainly dependent on various environmental exposures, antibiotic treatments, and genetics (*Bokulich et al., 2016*; *Azad et al., 2013*; *Jost et al., 2014*; *Palmer et al., 2007*; *Penders et al., 2013*).

Lately, studies employing metagenome sequencing have investigated strain-level transfer between mother and infants; *Asnicar et al., 2017* showed the feasibility of using metagenomics to identify shared bacterial strains between mother and infant, 3 months post-birth, in a handful of pairs, *Ferretti et al., 2018* showed that bacterial strains from mothers' stool appear more frequently in the infants' gut microbiota at a later age, and *Shao et al., 2019* showed that strain-level transmission of especially *Bacteroidetes* was stunted for CS-delivered children compared to vaginally delivered children. These studies all identify specific strains that show transfer between mother and infant. However, the high inter-individual variability in airway and fecal microbiomes, as well as the large differences when comparing them to the vaginal microbiome, makes it difficult to determine any statistical significance of such transfer. We therefore suggest a novel strategy for statistically testing for bacteria being consistently transferred from mother to infant, which is based on the calculation of weighted transfer ratios (WTRs) at higher taxonomic levels.

In this study, we investigated the vaginal microbiota during the last trimester of pregnancy and its importance for the development of the airway and fecal microbiota from early life up to age 3 months and 1 year, respectively. Transfer from mother to infant was assessed as WTR in general and at order level. We used vaginal samples from 665 pregnant women (gestational weeks 24 and 36) as well as airway (*Mortensen et al., 2016*) and fecal (*Stokholm et al., 2018*) samples from the children of 651 of these women, collected as part of the Copenhagen Prospective Studies on Asthma in Childhood 2010 (COPSAC$_{2010}$) cohort. The bacteria were identified by 16S rRNA gene amplicon sequencing.

## Results

### Vaginal microbiota

We successfully sequenced 1322 vaginal samples from gestational weeks 24 (n = 657) and 36 (n = 665), with a mean read count of 53,875 (interquartile range [IQR]: 38,330–64,632), representing 3287 unique Amplicon sequence variants (ASVs), with a mean observed richness of 25.0 (IQR: 12–32). We observed 28 unique phyla, of which 10 had a mean abundance above 0.1%, and the most abundant phyla were Firmicutes (85.2%), Actinobacteria (12.0%), and Proteobacteria (1.5%). We observed 463 genera, 94 with mean abundance above 0.1%, and dominated by Lactobacillus (81.1%) and Gardnerella (9.0%). Of the 3287 ASVs, 420 had a mean abundance above 0.1% and the most abundant ASVs were four Lactobacilli and one Gardnerella, with sequences matching *L. crispatus* (31.5%), *L. iners* (29.5%), *L. gasseri* (10.4%), *G. vaginalis* (4.6), and *L. jensenii* (4.5%). We clustered the vaginal samples into six clusters (*Supplementary file 1*, section 1.3.1), and based on the dominant ASVs in each cluster, we refer to them as CST I (*L. crispatus*), CST II (*L. gasseri*), CST III (*L. iners*), CST IV-a (*G. vaginalis*, few Lactobacillus), CST IV-b (*G. vaginalis*, some Lactobacillus), and CST V (*L. jensenii*), in accordance with *Gajer et al., 2012* (*Figure 1A*). The observed richness differed significantly between CSTs (p<10$^{-15}$, *Figure 1B*), being lowest in CSTs I, slightly higher in CST III and CST V, while CST IV-a and CST IV-b had the highest observed richness. Similarly, for Shannon diversity index, CST I and CST III were significantly lower than the rest, while CST IV-a and CST IV-b were also

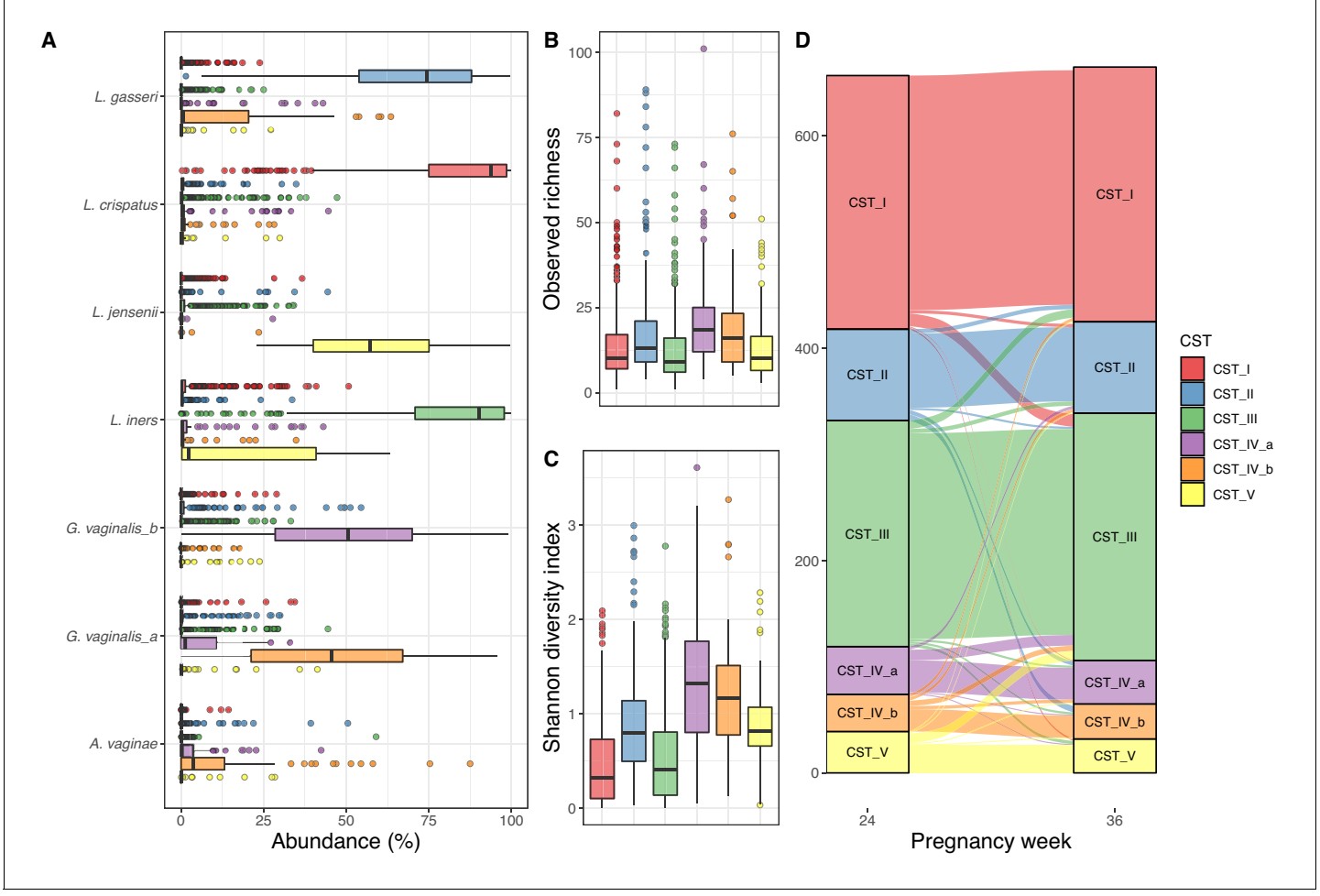

**Figure 1.** Vaginal community state type (CST). (**A**) Boxplot of top amplicon sequence variant (ASV) abundance for each CST (including two most abundant ASVs from each CST), (**B**) boxplot of observed richness by CST, (**C**) boxplot of Shannon diversity index by CST, and (**D**) alluvial plot showing the CST at weeks 24–36 for each woman. All plots are colored by the CST.

The online version of this article includes the following figure supplement(s) for figure 1:

**Figure supplement 1.** Non-metric multidimenionsal scaling plot based on Jensen–Shannon divergence; samples colored by community state type (CST), and gray lines connect samples from the individual women.

significantly higher than CST II and CST V (p<$10^{-15}$, *Figure 1C*, *Supplementary file 1*, section 1.3.2). We analyzed the stability of the vaginal microbiota in relation to their CSTs and by comparing intra- and inter-individual beta diversity distances.

Of the 657 women with both week 24 and 36 data, 562 (85.5%) had the same CST at week 36, as they had at week 24, with CST IV-b (21 of 35–60.0%) being significantly less stable ($\chi^2$ test, adjusted p-value<0.048), except CST IV-a (30 of 45–66.7%) and CST V (26 of 39–66.7%), while CST I and CST III were the most stable (92.5% and 91.1%, respectively) (*Figure 1*, *Supplementary file 1*, section 1.3.2.3.2). The median Jensen–Shannon divergence (JSD) between women's paired week 24 and week 36 samples (median$_{JSDpairs}$ = 0.031) was significantly lower than the median JSD between mismatched pairs of week 24 and week 36 samples (median$_{JSDnonpairs}$ = 0.635) (p<$10^{-3}$). Furthermore, the distance from week 24 to week 36 strongly depends on week 24 CST, with the highest median divergence for CST IV-b (median$_{JSDdist}$ = 0.093), followed by similar values for CST IV-a (median$_{JSDdist}$ = 0.076), CST V (median$_{JSDdist}$ = 0.068), and CST II (median$_{JSDdist}$ = 0.064), with the lowest median divergence for CST I (median$_{JSDdist}$ = 0.023) and CST III (median$_{JSDdist}$ = 0.022) (*Supplementary file 1*, section 1.3.2.3.2).

On a non-metric multidimensional scaling plot of JSD for the vaginal samples, it shows that CST I, CST II, and CST III are better defined than CST IV-a, CST IV-b, and CST V (*Figure 1—figure supplement 1*). A statistical test of the JSD dispersion confirmed that CST I and CST III were less dispersed than the other CSTs (adjusted $p<10^{-3}$), while both CST II and CST V were less dispersed than CST IV-a and CST IV-b (adjusted $p<0.02$). Lastly, the beta diversity was not dependent on sampling time point (PERMANOVA $p=0.86$), while CSTs were highly significant (PERMANOVA $R^2 = 0.80$, $p<0.001$, *Supplementary file 1*, section 1.4.2).

## Infant microbiota

The microbiota of the airway and fecal samples have previously been described in detail; for the full analysis, see *Mortensen et al., 2016* and *Stokholm et al., 2018* for the airway and fecal samples, respectively. Of the 695 children in the COPSAC$_{2010}$ cohort, we included the 651 children (94%) with a corresponding maternal week 36 vaginal sample. Of these children, 520 (79.8%) were delivered vaginally, 68 (10.5%) by in labor CS (CS-L), and 63 (9.7%) by scheduled CS (CS-S).

In the 1746 airway samples from the infants with a corresponding maternal week 36 sample (1 week: 526, 1 month: 606 and 3 months: 614), we identified 7500 ASVs, from 35 phyla, of which 8 had a relative abundance above 0.1%, dominated by Firmicutes (61%), Proteobacteria (30%), Actinobacteria (6%), and Bacteroidetes (2%). At genus level, we had 828 genera (37 above 0.1%), and the most dominant were *Staphylococcus* (25.6%), *Streptococcus* (25.6%), *Moraxella* (14.8%), and

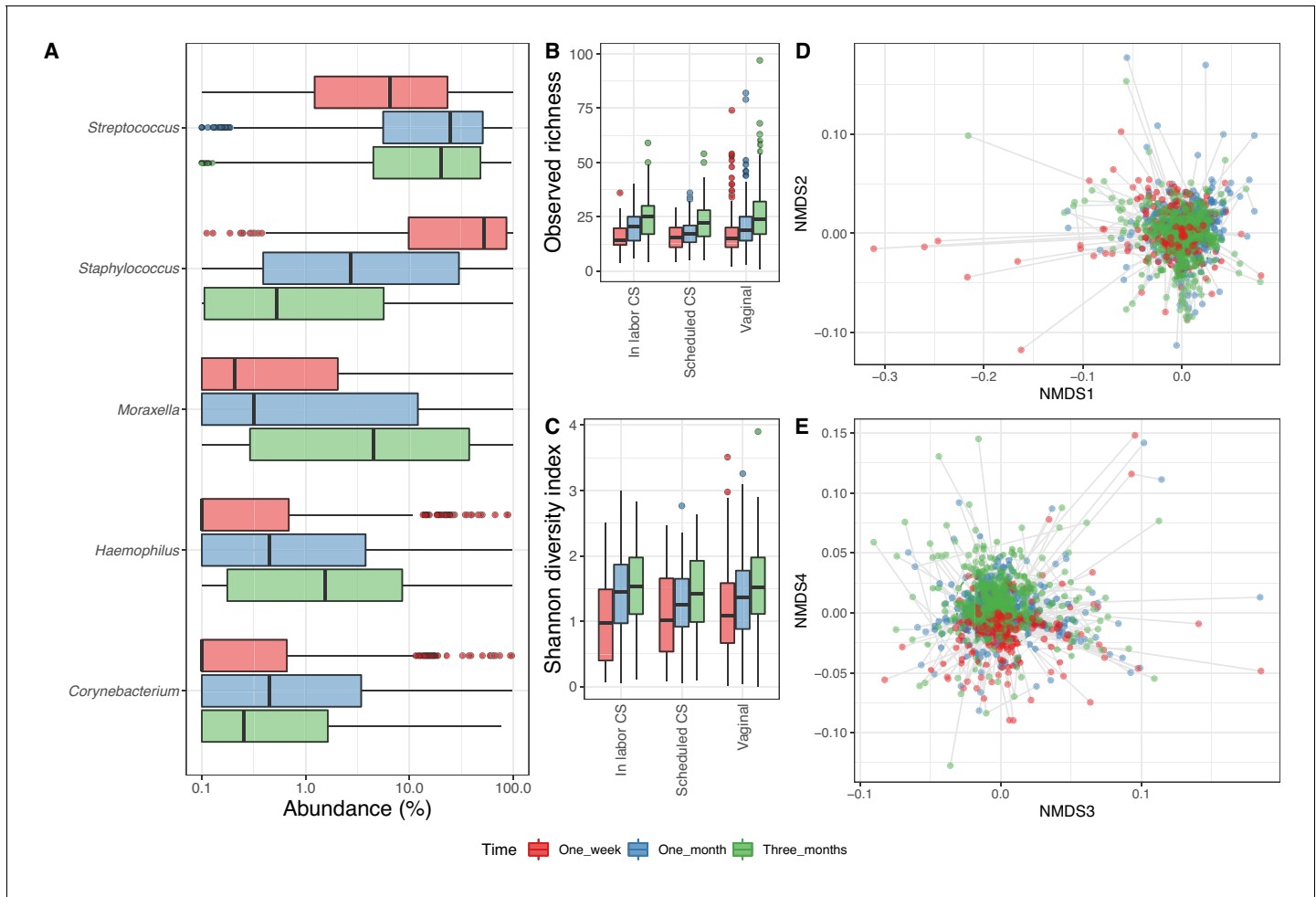

**Figure 2.** Airway microbiome. (A) Boxplot of top genera with mean abundance of 5% at least one time point. Abundance is plotted on a log10 scale, and all abundances below 0.1% have been set to 0.1% in the plot. (B–C) Boxplot of alpha diversity, (B) observed richness, and (C) Shannon diversity index, by delivery mode. Three samples with observed richness above 100 were excluded from the plot. (D–E) Non-metric multidimensional scaling plots based on weighted UniFrac distances. Samples from the same individual are connected by gray lines. All plots are colored by time point.

*Haemophilus* (6.0%) (**Figure 2A**). Of the 7500 ASVs, only 58 had a mean relative abundance above 0.1%, with the four most dominant being to Genus_Staphylococcus_205 (24.2%), Genus_Streptococcus_177 (18.4%), Genus_Moraxella_95 (13.7%), and Genus_Haemophilus_69 (3.5%) – see **Supplementary file 1**, section 2.2.2. The mean observed richness increased significantly from 1 week (16.6) to 1 month (20.5) and 3 months (25.1) (adjusted $p<10^{-15}$, **Figure 2B**) and was significantly lower for infants delivered by CS-S compared to vaginal delivered infants, when adjusting for time point (p=0.018). The Shannon diversity index also increased significantly over time (mean 1.1, 1.3, and 1.5 at 1 week, 1 month, and 3 months, respectively, adjusted $p<10^{-6}$, **Figure 2C**), with no observed differences between modes of delivery (**Supplementary file 1**, section 2.2.3). The time point significantly explained a small part of the variation in beta diversity (weighted UniFrac distances PERMANOVA, $R^2 = 0.013$, p<0.001, **Figure 2D–E**), while delivery method ($R^2 = 0.002$, p=0.97) and mother's CST at week 36 were not significantly associated ($R^2 = 0.004$, p=1).

In the 1688 fecal samples from the infants with a corresponding maternal week 36 sample (1 week: 533, 1 month: 575 and 1 year: 580), we identified 6818 ASVs. Thirty-three phyla were present in the fecal samples, of which five had a mean relative abundance above 0.1%, namely Bacteroidetes (34.4%), Proteobacteria (26.4%), Firmicutes (21.4%), Actinobacteria (16.2%), and Verrucomicrobia (1.4%). There were 707 genera in total, 44 with mean relative abundance above 0.1%, with the most abundant being *Bacteroides* (29.2%), *Bifidobacterium* (15.6%), and *Escherichia/Shigella* (14.1%) (**Figure 3A**). The fecal samples contained 6818 ASVs, 87 with mean relative abundance above 0.1%, and the most dominant being Genus_Escherichia_Shigella_101 (13.2%), Genus_Bifidobacterium_60 (10.0%), and Bacteroides_fragilis_22 (6.6%) (**Supplementary file 1**, section 2.3.2). The observed richness was similar at 1 week (23.2) and 1 month (22.1), before being more than twice as high at 1 year (53.1) ($p<10^{-15}$, **Figure 3B**). The same was reflected in the Shannon diversity index, which significantly increased from 1 week (1.5) and 1 month (1.4) to 1 year (2.3) (**Figure 3C**). For fecal samples, both alpha diversity measures were independent of delivery mode (p=0.26 and p=0.40 for observed richness and Shannon diversity index, respectively) (**Supplementary file 1**, section 2.3.3).

As for the airway samples, the time point significantly explained a small part of the variation in beta diversity (time point: $R^2 = 0.035$, p<0.001, **Figure 3D–E**), while delivery mode and mother's CST at week 36 alone were not significantly associated (p=1). Interestingly, when sequentially adding time points followed by delivery mode or mother's CST to the analysis, both did significantly explain an additional small fraction of the beta diversity (delivery mode: $R^2 = 0.004$, p<0.001, mother's CST: $R^2 = 0.007$, p<0.001) (**Supplementary file 1**, section 2.3.4).

## Transfer of the microbiota

Of a total of 3287 identified vaginal ASVs, for transfer according to compartment and time point, 293–404 were tested for vertical transfer in vaginally born children and 104–181 in CS-born children. These ASVs covered 31–66%, 36–90%, and 59–92% of the vaginal, airway, and fecal reads, respectively. **Table 1** shows summary statistics on the ASVs tested for each comparison.

### Transfer of specific ASVs

For each testable ASV, we calculated the odds ratio (OR) and p-value for transfer from mothers' vaginal microbiota to their child's fecal or airway microbiota at week 1, for both vaginally and CS-delivered infants. In general, only four ASVs (from the genera *Escherichia/Shigella*, *Koukoulia*, *Prevotella*, and *Ureaplasma*) showed false discovery rate (FDR)-corrected statistical significant transfer (**Supplementary file 1**, section 3.1.2.4). Of the four, none were significantly transferred at more than one combination of time point, delivery mode, and compartment, but the *Ureaplasma* ASV did tend to be more shared between mother and infant. Of the 92 samples from children containing this *Ureaplasma* ASV, 67 (70.7 %) of the matching samples from the mother shared the ASV (**Supplementary file 1**, section 3.1.2.3). When differentiating between CS-L (65–121 ASVs tested) and CS-S (58–107 ASVs tested), no ASVs were found to be significantly transferred after correcting for multiple testing (**Supplementary file 1**, section 3.1.2.3).

We then examined whether maternal abundances of the ASVs would affect the likelihood of transfer. By correlating the OR for transfer for each ASV with the population-wide relative maternal abundance, we revealed a negative correlation between maternal abundance of an ASV and the OR for transfer to the fecal compartment for vaginally (1 week: p=0.047, 1 month: p=0.0001, 1 year:

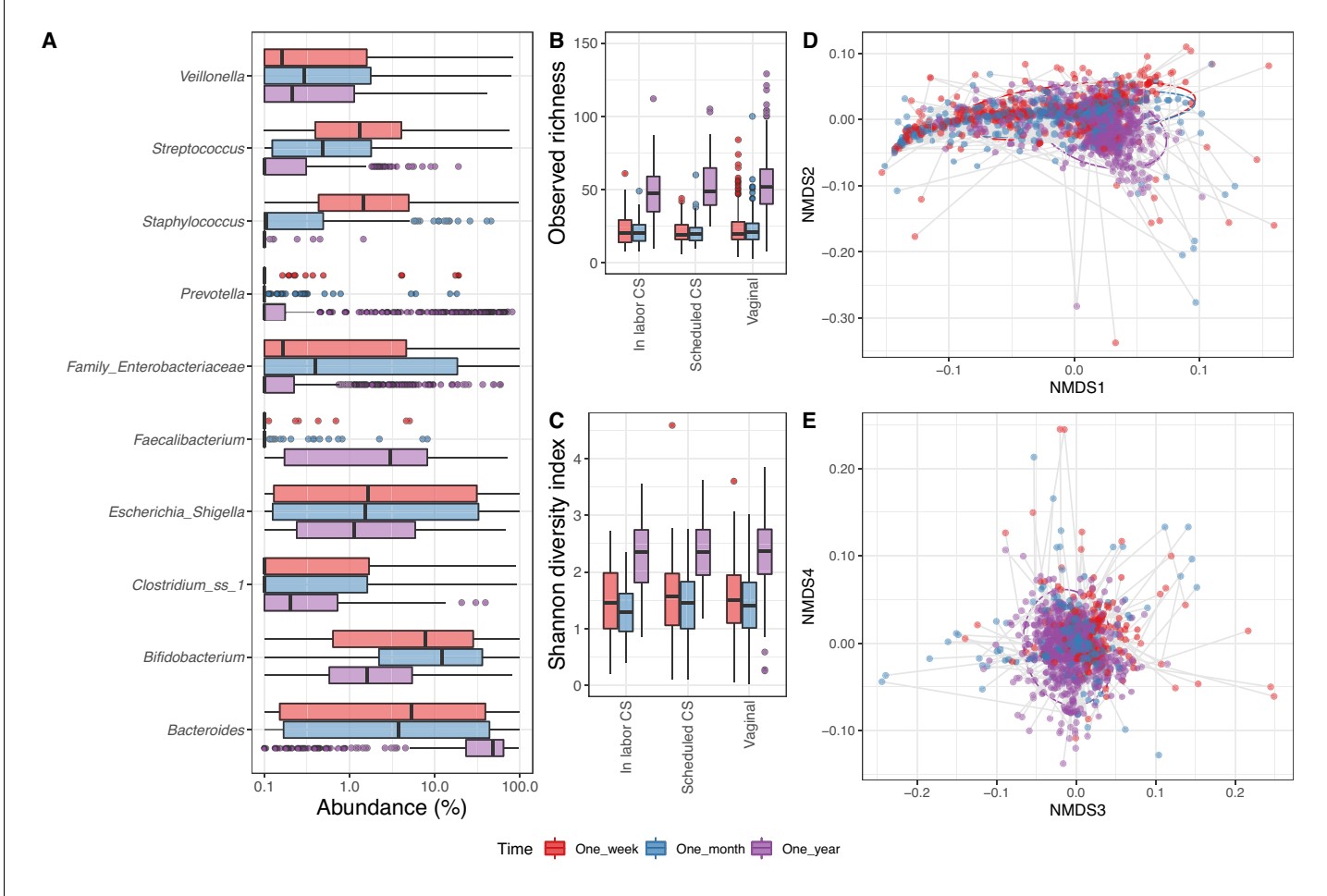

**Figure 3.** Fecal microbiome. (**A**) Boxplot of top genera with mean abundance of 4% at least one time point. Abundance is plotted on a log10 scale, and all abundances below 0.1% have been set to 0.1% in the plot. (**B–C**) Boxplot of alpha diversity, (**B**) observed richness, and (**C**) Shannon diversity index, by delivery mode. One sample with observed richness of >150 was excluded from the plot. (**D–E**) Non-metric multi-dimensional scaling plot based on weighted UniFrac distances. Samples from the same individual are connected by gray lines. All plots are colored by time point.

p=0.0024) and CS-born children at 1 year (p<0.015), whereas there was no association between population abundance and transfer odds to the airway (p>0.07). When differentiating between CS-L and CS-S, we did find a significant positive correlation for the airways at 1 week for CS-L (p=0.0012), while CS-S tended to have a negative correlation (p=0.055), and no significant correlations were identified for transfer odds to the fecal compartment (*Figure 4—figure supplements 1–4*, *Supplementary file 1*, section 3.1.3.5).

### Transfer of the most dominating vaginal ASV at week 36

For each dyad, the frequency of the most dominating maternal ASV was estimated in the children (*Supplementary file 1*, section 3.3.2). In the fecal compartment of vaginally delivered children, the proportion of children with their mother's most dominant ASV were 18–20% during the first year of life, while for CS-delivered children, the proportion decreased from 16% at 1 week to 13% at 1 year. In the airway compartment, there was a decrease over time, 13%, 12%, and 11% for vaginally born children and 9%, 8%, and 7% for CS-born children. A permutation test revealed that this association was only statistically significant for vaginally born children at 1 week for both compartments and also at 1 month for the airways (p<0.02), and for CS-born children at 1 week, the association was almost significant (p=0.07), while all other associations were far from significance (p>0.17).

**Table 1.** Descriptives on testable Amplicon sequence variant (ASV) in terms of numbers of ASVs, vaginal, fecal, and airway total coverage, number of tests reaching nominal, and false discovery rate-corrected significance.

| Compartment | Delivery mode | Age (days) | Testable ASVs (n) | Vaginal relative abundance (%) | Child relative abundance (%) | Min (p) * (n) | Min (q)† (n) | p<0.01 (n) | p<0.05 (n) | q<0.05 (n) | q<0.10 (n) |
|---|---|---|---|---|---|---|---|---|---|---|---|
| Airways | CS | 7 | 104 | 31.0 | 36.5 | 0.058 | 0.990 | 0 | 0 | 0 | 0 |
| Fecal | CS | 7 | 160 | 31.6 | 73.3 | 0 | 0.008 | 3 | 5 | 1 | 1 |
| Airways | CS | 30 | 131 | 56.4 | 85.0 | 0.008 | 0.347 | 3 | 6 | 0 | 0 |
| Fecal | CS | 30 | 181 | 60.5 | 83.2 | 0.008 | 0.991 | 1 | 4 | 0 | 0 |
| Airways | CS | 90 | 152 | 56.8 | 84.4 | 0.033 | 0.992 | 0 | 3 | 0 | 0 |
| Fecal | CS | 300 | 161 | 61.7 | 59.3 | 0.018 | 0.991 | 0 | 2 | 0 | 0 |
| Airways | V | 7 | 293 | 64.0 | 46.0 | 0.002 | 0.691 | 3 | 13 | 0 | 0 |
| Fecal | V | 7 | 354 | 65.2 | 90.7 | 0 | 0.012 | 12 | 28 | 2 | 4 |
| Airways | V | 30 | 342 | 63.9 | 90.2 | 0 | 0.001 | 8 | 14 | 1 | 2 |
| Fecal | V | 30 | 395 | 65.8 | 92.4 | 0.002 | 0.312 | 11 | 28 | 0 | 0 |
| Airways | V | 90 | 364 | 62.2 | 87.7 | 0.001 | 0.260 | 3 | 9 | 0 | 0 |
| Fecal | V | 300 | 404 | 64.2 | 84.5 | 0.003 | 0.457 | 7 | 17 | 0 | 0 |

*Uncorrected p-values.

†False discovery rate corrected using the Benjamini-Hochberg procedure.

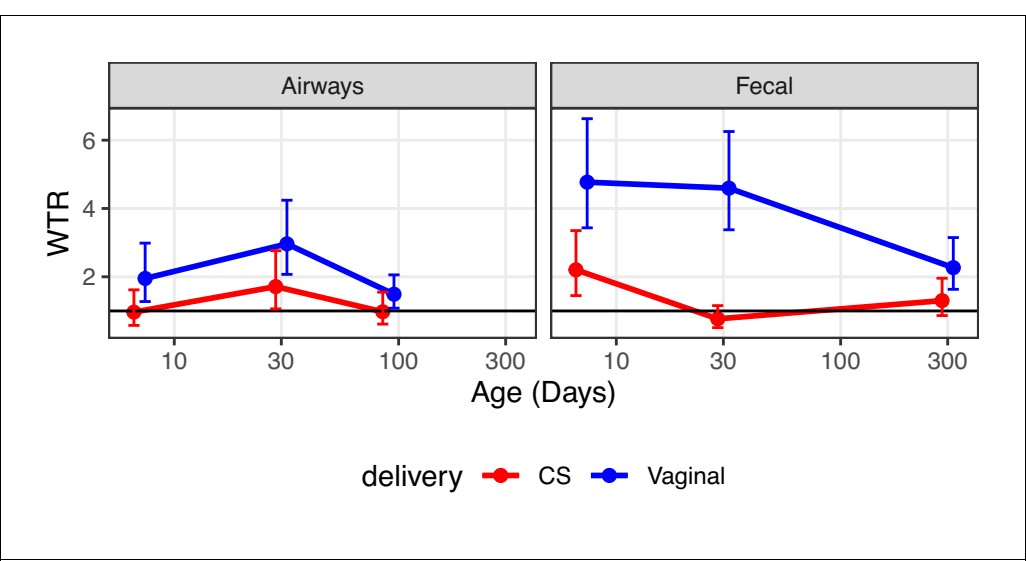

**Figure 4.** Weighted transfer ratios from vaginal week 36 to the fecal and airway compartments in first year of life stratified on mode of delivery (blue: vaginal birth, red: cesarean section). A ratio above one indicates enrichment of microbial transfer. Error bars reflect standard errors.

The online version of this article includes the following figure supplement(s) for figure 4:

**Figure supplement 1.** The odds for transfer between mother (week 36) and child for children delivered vaginally.

**Figure supplement 2.** The odds for transfer between mother (week 36) and child for children delivered by cesarean section (CS).

**Figure supplement 3.** The odds for transfer between mother (week 36) and child for children delivered by in labor CS.

**Figure supplement 4.** The odds for transfer between mother (week 36) and child for children delivered by scheduled CS.

## Enrichment of ASVs with positive transfer estimates (compared to negative)

In order to pursue an enrichment hypothesis, a WTR was calculated as function of compartment, birth mode, and age (*Figure 4*, *Supplementary file 1*, section 3.2.1). WTR is based on the OR and p-value for each individual ASV and calculated as the weighted ratio of positive OR (OR > 1) and negative OR (OR < 1), with WTR > 1 indicating enrichment and WTR < 1 indicating depletion of positive OR. For the fecal compartment, a clear enrichment was observed for vaginally born children, with attenuated strength over time ($WTR_{VAG}$: 4.9, 4.5, and 2.5, respectively. p<0.01). For CS-born children, enrichment was observed for the fecal compartment at the early time point ($WTR_{CS}$ = 2.6 for transfer, p=0.014), but not at the later time points ($WTR_{CS}$: 0.9 and 1.6, p>0.2, *Figure 4*). Interestingly, transfer to the airway compartment was significantly enriched at the first two time points for vaginally born children, but strongest at the age of 1 month (1 week: $WTR_{VAG}$ = 2.3, p=0.034, 1 month: $WTR_{VAG}$ = 3.4, p<0.001), with no significant enrichment observed at 1 year ($WTR_{VAG}$ = 1.6, p=0.094). For CS-born children, the WTR followed the same pattern, but at a lower level and without reaching significance at any time point ($WTR_{CS}$: 1.2, 2.1, 1.3, p>0.084, *Figure 4*). $WTR_{VAG}$ were higher than $WTR_{CS}$ at all time points, and compartments, but only significantly higher for transfer to the fecal compartment at 1 month (p=0.01, *Supplementary file 1*, section 3.2.1).

We performed a detailed enrichment analysis, in which we included the bacterial orders that had at least two testable ASVs for each combination of delivery mode, time point, and compartment, as well as one combination with more than 10 testable ASVs. The included orders were as follows: Clostridiales, Lactobacillales, Bacteroidales, Selenomonadales, Betaproteobacteriales, Pseudomonadales, Corynebacteriales, Bifidobacteriales, Enterobacteriales, and Bacillales. The analysis showed that the overall enrichment result was order dependent (*Figure 5*, *Supplementary file 1*, section 3.2.2.2). Several patterns were observed; some showed evidence of transfer, where positive and larger WTRs are observed for vaginally born children in comparison with CS-born children at 1 week, which were either transient (decreased over time) or persistent (maintained over time). Other patterns showed evidence of a common reservoir, where WTR increased in general and became more similar between mode of delivery with increasing age, indicating that bacteria were shared through a common living environment or transferred from mother to infant at a later time point (e.g. skin contact), and not transferred from mother's vaginal microbiome to child during birth.

Clostridiales, the most represented order, shows evidence of transfer from a common reservoir. For transfer to the fecal compartment, $WTR_{VAG}$ and $WTR_{CS}$ were almost identical, not showing transfer at 1 week or 1 month, but with enrichment at 1 year (p<0.02). While not statistically significant, there was evidence of transfer to the airway compartment at 1 week, with similar enrichment for vaginally and CS-born children at later time points, indicating sharing of ASVs after birth. For Lactobacillales, the WTRs for the airways were not significant at 1 week, but we did find significant $WTR_{VAG}$ (6.8, p=0.003) at 1 month, which persisted to 3 months (p=0.021), while $WTR_{CS}$ were similar to $WTR_{VAG}$ at 1 month, but decreased at 3 months. Both $WTR_{VAG}$ (13.9, p<0.001) and $WTR_{CS}$ (4.9, p=0.039) were significant for the fecal compartment at 1 week, and $WTR_{VAG}$ (17.5, p<0.001) at 1 month, but attenuated over the first year, indicating transient transfer. Bacteroidales were persistently transferred to the fecal compartment of vaginal born children, with $WTR_{VAG}$ being significant during the first month and attenuating slightly at 1 year, while $WTR_{CS}$ were not significant at any time point. For transfer to the airways, we did not have sufficient testable ASVs (n ≤ 15) to make a strong conclusion for CS-born children, and for vaginally born children, we did see significant $WTR_{VAG}$ at 1 month (10.4, p=0.006). Selenomonadales and Betaproteobacteriales both had significant $WTR_{VAG}$, to the fecal compartment, at 1 week, and also at 1 month and 1 year for Betaproteobacteriales, with indications of an attenuated, but consistent transfer at the following time points, while there were a low number of testable ASVs for CS-born children, $WTR_{CS}$ for Selenomonadales at 1 week (6.2, p=0.027) were the only $WTR_{CS}$, together this indicate transfer, which could be either persistent or transient. For transfer to the airways, we were limited by few testable ASVs for CS-born children, with no significant WTR observed at any time point independent of delivery mode. The remaining orders generally had few testable ASVs, leading to high standard error and no clear indication of transfer or a common reservoir, but we did observe significant $WTR_{VAG}$ for Enterobacteriales. For Enterobacteriales, $WTR_{VAG}$ to the fecal compartment were consistently significant and $WTR_{VAG}$ to the airway compartment were significant at 1 and 3 months, while $WTR_{CS}$ were high at

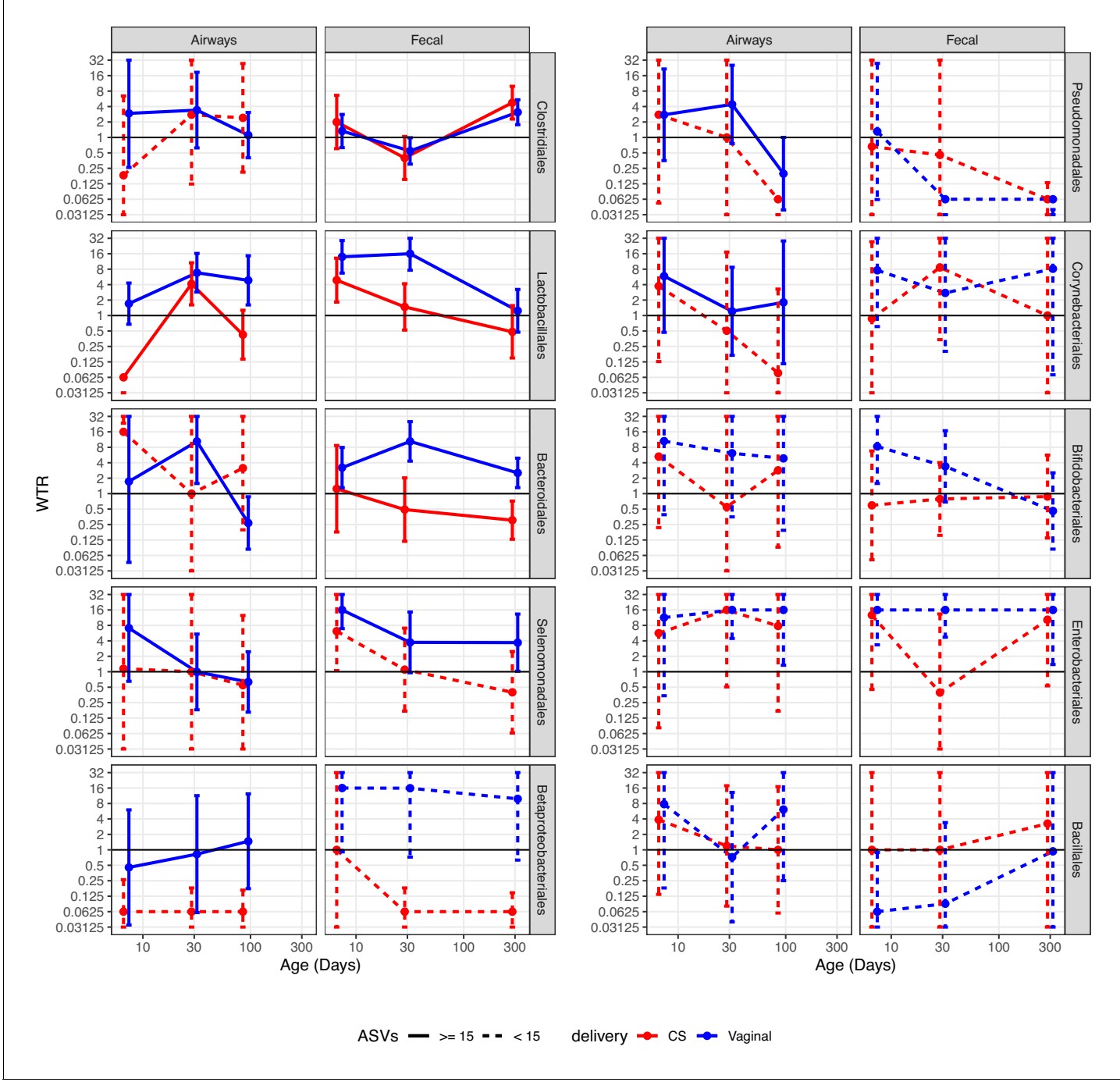

**Figure 5.** Weighted transfer ratios (WTRs) from vaginal week 36 to the fecal and airway compartments in the first year of life according to the mode of delivery (blue: vaginal birth, red: cesarean section) partitioned for the 10 most represented taxonomic classes at order level with upper left (Clostridiales) being the most represented order followed byLactobacillales and so forth. Dashed lines represent analysis on less than 15 ASVs on average. Error bars reflect standard errors. WTR is truncated so values lower than 0.625 are plotted as 0.625 and values higher than 16 are plotted as 16.

The online version of this article includes the following figure supplement(s) for figure 5:

**Figure supplement 1.** Weighted transfer ratios (WTRs) from vaginal week 36 to the fecal and airway compartments in the first year of life according to the mode of delivery (blue: vaginal birth, red: in labor CS, green: scheduled CS) partitioned for the 10 most represented taxonomic classes at order level with upper left (Clostridiales) being the most represented family followed by Lactobacillales and so forth.

1 week and 1 year, but with very large standard error, indicating that this could be persistent transfer.

When separating CS deliveries into CS-L and CS-S, for the orders with sufficient testable ASVs (Clostridiales, Lactobacillales, Bacteroidales, Selenomonadales, Betaproteobacteriales, and Enterobacteriales), we saw no larger differences compared to CS in general (*Figure 5—figure supplement 1*, *Supplementary file 1*, section 3.2.2.2). For Clostridiales, the WTRs for the airways were not significant, but $WTR_{CS-L}$ were more similar to vaginally delivered children at 1 and 3 months. WTRs in the fecal compartment were higher for CS-L-born children, and significant at 1 year, with $WTR_{CS-S}$ following the same trend, but lower. For Lactobacillales, $WTR_{CS-L}$ tended to be closer to $WTR_{VAG}$ than $WTR_{CS-S}$, but not by much. We did find a significant $WTR_{CS-L}$ for the airways at 1 month, but no significant WTR for the fecal compartment. For Bacteroidales, $WTR_{CS-L}$ were significant for the airways at 1 week, despite large variation between ASVs. For the fecal compartment, $WTR_{CS-L}$ followed the pattern of $WTR_{VAG}$, but lower and not significant. Surprisingly, Selemonadales had a significant $WTR_{CS-S}$ to the airways at 1 week, despite just three testable ASVs and large standard errors. For the fecal compartment, $WTR_{CS-L}$ were four times higher than $WTR_{CS-S}$ at all time points, and the significant $WTR_{CS}$ at 1 week seem to be driven by $WTR_{CS-L}$ being very similar to $WTR_{VAG}$. For Betaproteobacteriales, only a single WTR were higher than 1 ($WTR_{CS-S}$ to the fecal compartment at 1 week) but this WTR were calculated from just based on just three testable ASVs. For Enterobacteriales, there were few testable ASVs for CS-L and CS-S (2–7), and $WTR_{CS-L}$ to the fecal compartment at 1 week were the only significant WTR. Interestingly, for the airway compartment, $WTR_{CS-L}$ were almost identical to $WTR_{VAG}$ at all time points, while $WTR_{CS-S}$ consistently low. For the fecal compartment, $WTR_{CS-L}$ were four times higher than $WTR_{CS-S}$, and at $WTR_{VAG}$ level at 1 week and 1 year, but still showed the low WTR at 1 month that was also observed for $WTR_{CS}$.

## Discussion

The present study has characterized the vaginal microbiota of 665 Danish women during pregnancy and subsequently the microbiota of 651 children born by these women. In concordance with the previous smaller North American and European studies (*MacIntyre et al., 2015* [n = 42], *Ravel et al., 2013* [n = 396], *Romero et al., 2014* [n = 32+22]), we performed similar clustering of the vaginal samples and identified three clear clusters (CST I, CST II, and CST III), as well as three less-defined clusters, of which two were dominated by *Gardnerella* (CST IV-a and CST IV-b) and one by *Lactobacillus* (CST V). CSTs dominated by *Gardnerella* had higher alpha diversity and beta diversity dispersion than *Lactobacillus*-dominated CSTs, as seen in prior studies (*Ravel et al., 2011*; *Ma and Li, 2017*), while generally also being less likely to be stable between week 24 and week 36 of pregnancy. Based on the two vaginal sampling times, we saw a high intraindividual similarity, with the median beta diversity distance between each woman's own samples being 3.8–20.6 times smaller than the median distance to other women's samples, for weighted UniFrac distances and Jensen–Shannon divergence, respectively, and we found that 85.5% of all women belonged to the same CST at both time points. While our two sample points do not provide an in-depth proof of vaginal microbiota's stability, the high intraindividual stability was sufficient to consider the vaginal microbiota at week 36 as a relevant proxy for the vaginal bacteria that infants were exposed to during vaginal birth. We have not found correlations with children's airway microbiome, but the week 36 vaginal microbiota did explain a small fraction of the variation in beta diversity for the children's fecal microbiome.

Studies investigating transfer at strain level have shown that specific bacteria are transferred from mother to infant (*Asnicar et al., 2017*; *Ferretti et al., 2018*; *Shao et al., 2019*), from multiple maternal compartments, but none have included statistical test that confirms the significance of the observed transfer. While this study is based on amplicon sequencing and cannot achieve strain-level resolution, we analyzed transfer between 651 mother–child pairs, of 3287 vaginal ASVs, to two distinct compartments, across two modes of delivery, and three time points, which meant that 39,444 possible transfer models were considered. Of these, the transfer odds could be calculated for just 3041 models (7.7%) in total, with just four ASVs demonstrating significant transfer odds from mother to infant after correction for multiple testing, which highlight the differences between the vaginal, airway, and fecal microbiomes, as well as the inter-individual variation within either compartment. We found a negative correlation between ASV mean maternal abundance and transfer odds.

Interestingly, when disregarding the identity of the ASVs and instead rank the ASVs by maternal abundance within each dyad to calculate the transfer odds for the top ranked ASVs, we found that most abundant ASVs in each dyad were significantly transferred to the child's airway and fecal compartments at 1 week, for vaginally delivered children only.

These results indicate that another statistical approach is needed to better test whether bacteria are transferred from mother to child in a consistent and statistical significant manner. We developed the concept of WTR, an enrichment type of analysis, and applied it to better understand microbiome transfer at birth. This analysis indicated an inflated number of positive transfer odds at 1 week of age (4.7 times more for fecal samples and 1.9 times more for airway samples), which attenuated in strength from 1 month to 1 year of life.

Analyzing WTR at order level showed very different patterns for the tested orders, showing indications of transfer after birth, transient transfer, and persistent transfer. For transfer to the fecal compartment, the strongest indication of transfer after birth was for Clostridiales, with nearly identical WTR$_{VAG}$ and WTR$_{CS}$ around one at the two early time points and above three at 1 year. The signature of transient transfer was observed for Lactobacillales and Bifidobacteriales to the fecal compartment, where there was a strong enrichment of transfer odds for Lactobacillales at 1 week and 1 month, attenuated over time. For Bifidobacteriales, WTR$_{VAG}$ were significant at 1 week and sharply attenuating over time, which is very similar to the pattern found by strain-level analysis (*Asnicar et al., 2017*; *Shao et al., 2019*). Bacteroidetes had consistently enriched WTR$_{VAG}$, with no WTR$_{CS}$ being significant, strongly indicating persistent transfer of Bacteroidetes. While based on fewer ASVs (10–15), Enterobacteriales were also found to be persistently transferred to vaginally delivered children only. Interestingly, the four orders presenting patterns of WTR that indicate either persistent or transient transfer all contain bacteria that have been identified as part of the shared microbiota between breast milk, mothers' fecal samples, and infants' fecal samples (*Jost et al., 2014*), indicating that microbiota transfer after delivery may be more important than during delivery itself. This is further supported by a study showing that almost 40% of infants' fecal microbiota in the first 30 days of life originated from either breast milk or areolar skin microbiota (*Pannaraj et al., 2017*).

For transfer to the airways, only Enterobacteriales and Lactobacillales had a significant WTR$_{VAG}$, both at 1 month and at 1 year, with no WTR$_{CS}$ being significant for any order. This could indicate that even this approach does not have the power to compensate for the inter-individual differences in the airway microbiome.

When comparing vaginally delivered children to CS-delivered children, it is important to consider whether it was a CS-L or CS-S. A CS-L is performed if there are complications (such as arrest of descent) during a vaginal delivery and are likely to include lengthy exposure to the vaginal microbiome, while CS-S is done before delivery has begun and will be a more sterile and controlled procedure where the newborn is not exposed to the vaginal microbiome. We calculated transfer odds for CS-L (65–121 ASVs) and CS-S (58–107 ASVs) separately and were able to calculate WTR for six orders (Clostridiales, Lactobacillales, Bacteroidales, Selenomonadales, Betaproteobacteriales, and Enterobacteriales). While the stratification lowers statistical power, we did find WTR$_{CS-L}$ for CS-L to be less different from WTR$_{VAG}$ than WTR$_{CS-S}$ for CS-S, which correlates well with CS-L being more similar to vaginal delivery than CS-S, both when considering the child's early microbiome (*Stokholm et al., 2016*; *Chu et al., 2017*) and risk of developing asthma (*Sevelsted et al., 2016*).

Many studies have shown that bacteria transferred from mother to infant can originate from a range of maternal body sites, including, but not limited to, vagina, gut, and skin (*Dominguez-Bello et al., 2016*; *Chu et al., 2017*; *Ferretti et al., 2018*), with *Ferretti et al., 2018* identifying mother's stool as the most dominant source of transmission to infants' fecal microbiome. Additionally, *Rasmussen et al., 2020* identified specific ecological succession in the vaginal microbiota during pregnancy and birth, leading to decreased abundance of *Lactobacillus* and higher abundance of most other taxa at birth, while phylogenetically different microbes more commonly associated with other microbial compartments, such as gut, airways, and skin, were introduced. With that in mind, we acknowledge that this study does have limitations and cannot show the complete picture of maternal–infant microbiome transfer. Having only vaginal samples from the mothers it is clear that this study is not designed to refute maternal to infant transfer, as any lack of transfer can be attributed to not having sampled other maternal compartments. Despite such limitations, we could still

apply the concept of WTR to identify bacterial orders for which the WTR strongly suggests that transfer of maternal vaginal bacteria occurs during vaginal delivery.

In summary, we have shown that the vaginal microbiome in samples collected at week 24 and week 36 of pregnancy from pregnant women is highly correlated, indicating a stable community. With the assumption that the vaginal microbiome present at week 36 is indicative of the vaginal microbiome at birth, we found no overall correlation with the airway microbiome during the first 3 months after birth and only very small correlations with the fecal microbiome during the first year of life. Stratifying for time point and delivery mode to calculate the transfer odds for individual ASVs, we found minimal evidence of ASV-specific transfer from mothers' vaginal microbiome. To assess whether bacterial taxa were consistently transferred, we introduce the concept of weighted transfer ratios, and by comparing the development of WTR, over time, between delivery modes, we can identify patterns that suggest evidence of orders that are persistently or transiently transferred during delivery, as well as orders that are shared between mother and child at a later time point.

## Materials and methods

### Study population

COPSAC$_{2010}$ is an ongoing Danish mother–child cohort study of 700 unselected children and their families followed prospectively from pregnancy week 24 in a protocol previously described (*Bisgaard et al., 2013*). Exclusion criteria were as follows: gestational age below week 26; maternal daily intake of more than 600 IU vitamin D during pregnancy; or having any endocrine, heart, or kidney disorders.

### Sample collection

Vaginal samples from women at gestational weeks 24 and 36 were collected from the posterior fornix of the vagina using flocked swabs (ESWAB regular, SSI Diagnostica, Hillerød, Denmark) (*Stokholm et al., 2012*). Airway samples were aspirated with a soft suction catheter passed through the nose into the hypopharynx as previously described in detail (*Bisgaard et al., 2007*). Fecal samples were collected in sterile plastic containers and transported (within 24 hr) to Statens Serum Institute (Copenhagen, Denmark). Each sample was mixed on arrival with 10% vol/vol glycerol broth (SSI, Copenhagen, Denmark) and frozen at −80°C until further processing (*Stokholm et al., 2018*). Two thousand six hundred and seventy samples were collected and initially included.

The airway microbiota samples used in this study have been presented previously in *Mortensen et al., 2016* , and the fecal samples have been presented in *Stokholm et al., 2018*. For both sample types, three consecutive samples were included to investigate transfer from mother to infant: from feces at 1 week, 1 month, and 1 year, and from airways at 1 week, 1 month, and 3 months.

### DNA extraction

Genomic DNA was extracted from the mothers' and infants' samples as described in *Mortensen et al., 2016*, using the PowerMag Soil DNA Isolation Kit optimized for epMotion (MO-BIO Laboratories, Inc, Carlsberg, CA) using the epMotion robotic platform model (EpMotion 5075VT, Eppendorf, Hamburg, Germany).

### 16S Amplicon sequencing and bioinformatics pipeline

16S rRNA gene amplification was performed as described in *Stokholm et al., 2018*, using a two-step PCR method, targeting the hypervariable V4 region (forward primer 515F: 5′-GTGCCAGC MGCCGCGGTAA-3′ [*Turner et al., 1999*], reverse primer Uni806R: 5′-GGACTACHVGGGTWTCTAA T-3′ [*Takai and Horikoshi, 2000*]). Amplicon products were purified with Agencourt AMPure XP Beads (Beckman Coulter Genomics, MA) and the purified products quantified with Quant-iT Pico-Green quantification system (Life Technologies, CA) to allow for pooling, in equimolar concentration, of up to 192 samples per library. The pooled DNA samples were concentrated using the DNA Clean and Concentrator-5 Kit (Zymo Research, Irvine, CA) and quantified again. The libraries were sequenced on the Illumina MiSeq System (Illumina Inc, CA) using MiSeq Reagent Kits v2.

Primers were removed from the raw paired-end FASTQ files generated via MiSeq using 'cutadapt' (*Martin, 2011*). Furthermore, reads were analyzed by QIIME2 (*Bolyen et al., 2019*) (qiime2-2018.11) pipeline, with forward and reverse reads truncated at 180 bp and 160 bp, through dada2 (*Callahan et al., 2016*) to infer the ASVs present and their relative abundances across the samples. Taxonomy was assigned using a pre-trained Naïve Bayes classifier (Silva database, release 132, 99% ASV) (*Quast et al., 2013*). ASV identifiers were created from their species, or lowest taxonomical classification, with an additional integer for ASVs with identical classification.

## Bioinformatics analysis

For data treatment and analysis, we used the open source statistical program 'R' (*R Development Core Team, 2020*), predominantly the R-package 'phyloseq' (*McMurdie and Holmes, 2013*), with the complete analysis contained as an Rmarkdown file (*Source code 1*). Samples with less than 2000 sequences were excluded. Two thousand three hundred and fifty-nine samples were included containing, on average, over 32,000 sequences per sample, representing 3934 distinct ASVs. JSD was used to describe the beta diversity in the sample set. As this method is sensitive to bias due to sequencing depth, we performed the calculation of JSD on a randomly subsampled ASV table with an even sequencing depth of 2000 observations. No other analysis was performed using the subsampled ASV table.

## Clustering analysis

Clustering analysis was performed using partitioning around medoids clustering, based on JSD, and the optimal number of clusters was chosen based on multiple cluster validation techniques using the R-package 'fpc' (*Hennig, 2020*): average silhouette width (*Rousseeuw, 1987*), Pearson gamma index (*Halkidi et al., 2001*), dunn2 (*Dunn†, 1974*), Caliñski and Harabasz index (*Caliñski and Harabasz, 1974*), as well as comparison with similarity to CST presented in prior studies.

## Identification of CST-dominant ASVs

For the three most dominant ASVs in each vaginal CST, we performed BLASTN (v 2.10.1+) against the NCBI 16S ribosomal RNA database (*Zhang et al., 2000*), excluding models (XM/XP) and uncultured/environmental samples (*Source data 2*). Alignments were filtered to minimum 99% identity, and the ASVs were assigned to unique species, when possible. In these cases where multiple species aligned equally well, we used published studies to select the most likely species. As an example, *L. crispatus* and *L. acidophilus* could not be distinguished based on the sequenced region, and as published studies on the vaginal microbiota concur that *L. crispatus*, in contrast to *L. acidophilus*, constitute an important part of the vaginal microbiota, we refer to *L. crispatus*/*L. acidophilus* solely as *L. crispatus*. Based on this reasoning, we will also refer to *L. gasseri*/*johnsonii* solely as *L. gasseri* (*Antonio et al., 1999*; *Kiss et al., 2007*; *Vásquez et al., 2002*).

## Stability of vaginal microbiota

The differences in the amount of women with a stable or non-stable CST were assessed using the $\chi^2$ test. The median JSD between each woman's paired samples tested against 2500 permutations of random unpaired sample sets.

## Transfer of microbiota

Analysis of transfer of microbiota was pursued by three approaches, namely (1) transfer of specific ASVs, (2) transfer of the most dominating vaginal ASV at week 36, and (3) enrichment of ASVs with positive transfer estimates (compared to negative). For all approaches, we analyzed transfer from vagina week 36 to both airways and gut, for all time points, in order to evaluate the differences between vaginally and CS-delivered children. To determine whether the microbiota was transferred from mother to infant, we used a Fisher's exact test comparing the presence/absence of ASVs between dyads of mothers and children and recorded the OR for transfer with a one-sided p-value toward the null hypothesis of OR = 1. Only ASVs showing the presence/absence in both the vaginal and the child compartments were included in the analysis. This comparison was conducted for both microbial compartments at all three time points during the first year of life and stratified by vaginal

and CS deliveries summing up to a total of 12 models. Inference for transfer of single ASVs was evaluated using Benjamini-Hochberg FDR correction.

## Weighted transfer ratio

A weighted transfer ratio between positive and negative ORs was used as an overall measure of transfer. WTR is defined as follows:

$$WTP = \frac{WP}{WN}$$

where *WP* and *WN* are the sum of the areas of the positive and negative associations from the volcano plot, respectively, that is

$$WP = \sum_{i \in I(OR>1)} -\log(OR_i)\log10(pv_i)$$

and

$$WN = \sum_{i \in I(OR<1)} \log(OR_i)\log10(pv_i)$$

giving larger emphasis to ASVs with high inference and effect size. $OR_i$ and *p.value*$_i$ refer to the OR and its corresponding null hypothesis test, respectively, for the *i*'th ASV. WTR should be around one in case of no transfer, and larger when present, but due to the high sparsity, the null distribution is not always centered on 1. To test for transfer, the dyads are scrambled to construct a null distribution for the ratio and the WTR reported is the model ratio relative to the median of the null distribution.

## Declarations

### Ethics approval and consent to participate

This study followed the principles of the Declaration of Helsinki, and was approved by the Ethics Committee for Copenhagen (The Danish National Committee on Health Research Ethics) (H-B-2008–093) and the Danish Data Protection Agency (2008-41-2599). Written informed consent was obtained from both parents for all participants. The study is reported in accordance with the Strengthening the Reporting of Observational Studies in Epidemiology (STROBE) guidelines (*von Elm et al., 2007*).

### Consent for publication

Not applicable.

## Acknowledgements

We express our deepest gratitude to the children and families of the COPSAC$_{2010}$ cohort study for all their support and commitment. We acknowledge and appreciate the unique efforts of the COPSAC research team. We thank Susanne Schjørring for critical logistic support, with handling and storing of samples at Statens Serum Institut, Denmark. Furthermore, we thank Karin Pinholt Vestberg, April Cockburn, and Jakob Russel (Section for Microbiology, University of Copenhagen) for the help and support with DNA extraction, construction of the 16S rRNA gene amplicon libraries, and sequencing.

## Additional information

### Funding

| Funder | Grant reference number | Author |
| --- | --- | --- |
| Lundbeckfonden | R93-A8944 | Hans Bisgaard<br>Søren Johannes Sørensen |

| Lundbeckfonden | R16-A1694 | Hans Bisgaard |
|---|---|---|
| Danish Ministry of Health | 903516 | Hans Bisgaard |
| Strategiske Forskningsråd | 0603-00280B | Hans Bisgaard |
| The Capital Research Foundation | | Hans Bisgaard |

The funders had no role in study design, data collection and interpretation, or the decision to submit the work for publication.

## Author contributions

Martin Steen Mortensen, Formal analysis, Validation, Investigation, Visualization, Writing - original draft, Writing - review and editing; Morten Arendt Rasmussen, Data curation, Software, Validation, Visualization, Writing - original draft, Writing - review and editing; Jakob Stokholm, Resources, Formal analysis, Validation, Investigation, Methodology, Writing - review and editing; Asker Daniel Brejnrod, Data curation, Software, Formal analysis, Validation, Writing - review and editing; Christina Balle, Investigation, Writing - review and editing; Jonathan Thorsen, Data curation, Formal analysis, Validation, Visualization, Writing - review and editing; Karen Angeliki Krogfelt, Conceptualization, Resources, Funding acquisition, Methodology, Project administration, Writing - review and editing; Hans Bisgaard, Søren Johannes Sørensen, Conceptualization, Resources, Supervision, Funding acquisition, Methodology, Project administration, Writing - review and editing

## Author ORCIDs

Martin Steen Mortensen https://orcid.org/0000-0001-5483-7533
Morten Arendt Rasmussen https://orcid.org/0000-0001-7431-5206
Jakob Stokholm https://orcid.org/0000-0003-4989-9769
Asker Daniel Brejnrod https://orcid.org/0000-0002-1327-2051
Christina Balle http://orcid.org/0000-0003-3439-3992
Jonathan Thorsen https://orcid.org/0000-0003-0200-0461
Karen Angeliki Krogfelt http://orcid.org/0000-0001-7536-3453
Hans Bisgaard https://orcid.org/0000-0003-4131-7592
Søren Johannes Sørensen https://orcid.org/0000-0001-6227-9906

## Ethics

Human subjects: This study followed the principles of the Declaration of Helsinki, and was approved by the Ethics Committee for Copenhagen (The Danish National Committee on Health Research Ethics) (H-B-2008-093) and the Danish Data Protection Agency (2008-41-2599). Written informed consent was obtained from both parents for all participants. The study is reported in accordance with the Strengthening the Reporting of Observational Studies in Epidemiology (STROBE) guidelines.

## Decision letter and Author response

Decision letter https://doi.org/10.7554/eLife.57051.sa1
Author response https://doi.org/10.7554/eLife.57051.sa2

# Additional files

## Supplementary files

• Source code 1. Rmarkdown file (FullAnalysis.Rmd) with the entire analysis presented here as well as the scripts necessary to recreate the entire analysis (getTransferStats.R, transferFunctions.R, getWinnerStats.R, and inferenceTransferStat.R).

• Source data 1. All tables produced in the analysis for this study.

• Source data 2. Results of the BLAST analyses performed in this study.

• Supplementary file 1. All results, including figures and tables, created as part of the analysis for this study.

• Transparent reporting form

## Data availability

The raw sequencing data analysed in this study is available in the Sequence Read Archive (SRA) repository under BioProject accession PRJNA691357. The prepared phyloseq object necessary to recreate the analyses presented in this study and data files with results from computational intensive analyses can be downloaded following links in FullAnalysis.Rmd (in Source code 1).

The following dataset was generated:

| Author(s) | Year | Dataset title | Dataset URL | Database and Identifier |
|---|---|---|---|---|
| Mortensen MS, Rasmussen MA, Stokholm J, Brejnrod AD, Balle C, Thorsen J, Krogfelt KA, Bisgaard H, Sørensen SrJ | 2020 | Maternal - infant microbiome transfer (COPSAC2010 cohort) | https://www.ncbi.nlm.nih.gov/bioproject/?term=PRJNA691357 | NCBI BioProject, PRJNA691357 |

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
