## [Decision Letter]

**Acceptance summary:**

Mortensen, Rasmussen et al. show that the vaginal microbiota is relatively stable and identify bacteria that show patterns of transmission from mother to child. This paper provides a useful new approach for modeling these transmission events that could be broadly applicable to microbiome studies and identify key bacteria for follow-on mechanistic work.

**Decision letter after peer review:**

Thank you for submitting your article "Stability of the vaginal microbiota during pregnancy and its importance for early infant colonization" for consideration by *eLife*. Your article has been reviewed by three peer reviewers, one of whom is a member of our Board of Reviewing Editors, and the evaluation has been overseen by Michael Eisen as the Senior Editor. The following individual involved in review of your submission has agreed to reveal their identity: Kjersti Aagard (Reviewer #2).

The reviewers have discussed the reviews with one another and the Reviewing Editor has drafted this decision to help you prepare a revised submission.

Summary:

The manuscript by Mortensen, Rasmussen et al. describes the analysis of the maternal vaginal microbiome paired with their children's fecal and airway microbiota. The authors used OTU-clustered V4 16S rRNA amplicon sequencing to compare 4990 samples from ~700 mother-child pairs. Using community state types (the vaginal microbiome field's equivalent to gut enterotypes), the authors demonstrated that the vaginal microbiota was relatively stable and conducted analysis looking at the transfer of features from mother to child finding relatively few taxa which transfer despite the relatively low resolution of amplicon sequencing which I would anticipate would inflate this estimate. The authors should be commended for their transparent analysis and the associated supplementary file detailing the source code for their analysis.

There are multiple strengths to this manuscript, including the large cohort size (657 women with two time point samplings) and the up to 1 year follow up data. Additional strengths include the measurement of both infant stool and airways microbiomes, which yielded some novel and interesting findings. Finally, we found their approach at transfer estimates interesting and the notion of using a weighted transfer ratio both novel and worthy of consideration by the scientific community at large.

While the reviewers all agreed that there are positive aspects to this study, substantially more data would be required prior to resubmission, including the addition of amplicon sequence variant analysis using the current 16S rRNA gene sequencing data, metagenomic sequencing on a subset of samples, as well as additional bioinformatics on the current dataset.

Essential revisions:

1) 16S amplicon sequencing is not a great method for evaluating bacterial transmission due to its low resolution. The authors should re-sequence at least a subset of their samples using metagenomic shotgun sequencing, enabling strain-level analyses, as in other recent studies of the early life microbiome (e.g., PMID: 25974306).

2) Given the specificity sought, it would be preferable if the authors were to use denoised sequence variants rather than OTUs which may inflate estimates of transfer and/or avoid the use of shallow subsampling which may decrease transfer estimates. The manuscript would also benefit from more in-depth comparison to published metagenomic data looking at strain transfer some of which draw differing conclusions (examples: Asnicar et al., 2017, Ferretti et al., 2018, Nayfach et al. Genome Research 2016). Using this published data to corroborate the taxonomic descriptions made would help support the findings.

3) The transmission analysis, while creative, lacks validation and seems overly simplistic. The first analysis, to our understanding, is testing which OTUs are consistently transferred across all mother-child pairs, which is a different question that asking which OTUs are transferred at all. Given the massive inter-individual variation in the gut microbiota at the OTU level it's not at all surprising to me that this analysis would give a low number. The second analysis is interesting but based upon the first estimate, which seems flawed to me. The third analysis doesn't stand up to much scrutiny – why would the most abundant OTU in the vagina be more likely to be transferred? Presumably, the dominant OTUs are adapted to the vagina and thus will not thrive as well in the gut or airway. As is, both figures are based upon these metrics that are highly questionable, limiting the clear conclusions that can be drawn from this study.

4) Superficial data analysis. Simply stating the average relative abundance of different taxa isn't enough by today's standards. More figures that show less derived statistics should be added in accordance with other papers in this field; for example, plots of relative abundance, comparisons of alpha/beta diversity metrics, etc. As is, the paper jumps directly to the transmission statistics without any of the typical analyses found in microbiome studies.

5) Avoid causal claims. The impact statement, title, background, and conclusion all make causal claims about factors that shape early life establishment of the gut microbiota despite the lack of causal support in humans.

---

## [Author Response]

Essential revisions:1) 16S amplicon sequencing is not a great method for evaluating bacterial transmission due to its low resolution. The authors should re-sequence at least a subset of their samples using metagenomic shotgun sequencing, enabling strain-level analyses, as in other recent studies of the early life microbiome (e.g., PMID: 25974306).

This is unfortunately not possible due to the lack of additional sample material for sequencing.

2) Given the specificity sought, it would be preferable if the authors were to use denoised sequence variants rather than OTUs which may inflate estimates of transfer and/or avoid the use of shallow subsampling which may decrease transfer estimates. The manuscript would also benefit from more in-depth comparison to published metagenomic data looking at strain transfer some of which draw differing conclusions (exs: Asnicar et al., 2017, Ferretti et al., 2018, Nayfach et al. Genome Research 2016). Using this published data to corroborate the taxonomic descriptions made would help support the findings.

We have now updated the dataset to denoised ASVs. This indeed reduces the amount of reads, but in a way that we assume adds more specificity to the observations. Consequently, the transfer analysis, comparing individual ASV transfer odds at family level is limited to families with a sufficient number of ASVs to actually estimate a meaningful weighted transfer ratio. However, the overall transfer result settles as the same conclusions as the OTU based analysis. The entire manuscript has been updated accordingly.

3) The transmission analysis, while creative, lacks validation and seems overly simplistic. The first analysis, to our understanding, is testing which OTUs are consistently transferred across all mother-child pairs, which is a different question that asking which OTUs are transferred at all. Given the massive inter-individual variation in the gut microbiota at the OTU level it's not at all surprising to me that this analysis would give a low number. The second analysis is interesting but based upon the first estimate, which seems flawed to me. The third analysis doesn't stand up to much scrutiny – why would the most abundant OTU in the vagina be more likely to be transferred? Presumably, the dominant OTUs are adapted to the vagina and thus will not thrive as well in the gut or airway. As is, both figures are based upon these metrics that are highly questionable, limiting the clear conclusions that can be drawn from this study.

Without testing for consistent transfer, any shared OTU (now ASV) will just be anecdotal, without statistical support. We would argue that just looking for shared ASV is too simplistic and that we here use a method that provides actual statistical evidence of transfer. In terms of validation, we have included permutation testing explicitly scrambling the mother-child relation, and hence in this case estimate the null transfer. Given distinct results based on mode of delivery further strengthen the validity of the results, as the early microbiome has been shown in numerous studies to depend on this factor. We have collaborated with O’Hely et al. and they have now submitted a similar analysis of transfer, using the same method, to *eLife* as supportive evidence of the validity of the method.

4) Superficial data analysis. Simply stating the average relative abundance of different taxa isn't enough by today's standards. More figures that show less derived statistics should be added in accordance with other papers in this field; for example, plots of relative abundance, comparisons of alpha/beta diversity metrics, etc. As is, the paper jumps directly to the transmission statistics without any of the typical analyses found in microbiome studies.

Our samples from each body site are either comparable to similar samples described in the field or they have already been described in other manuscripts (ref: Mortensen et al., 2016, Stokholm et al., 2018 (doi.org/10.1038/S41467-017-02573-2)). We agree with the reviewer that this should be added to make the current work coherent with other studies. We have therefore expanded our description of the infant microbiome composition and included the suggested comparisons of alpha and beta diversity metrics (Results).

5) Avoid causal claims. The impact statement, title, background, and conclusion all make causal claims about factors that shape early life establishment of the gut microbiota despite the lack of causal support in humans.

We agree. As part of the complete rewrite of this manuscript, we have attempted to remove any unsupported causal claims.